# Metoprolol and *CYP2D6*: A Retrospective Cohort Study Evaluating Genotype-Based Outcomes

**DOI:** 10.3390/jpm13030416

**Published:** 2023-02-26

**Authors:** Savannah Collett, Amanda Massmann, Natasha J. Petry, Joel Van Heukelom, April Schultz, Tadd Hellwig, Jordan F. Baye

**Affiliations:** 1Department of Pharmaceutical Services, Sanford USD Medical Center, Sioux Falls, SD 57117, USA; 2Roger Maris Cancer Center, Sanford Medical Center, Fargo, ND 58102, USA; 3Sanford Imagenetics, Sanford Health, Sioux Falls, SD 57105, USA; 4Sanford School of Medicine, University of South Dakota, Vermillion, SD 57069, USA; 5Department of Pharmacy Practice, North Dakota State University, Fargo, ND 58108, USA; 6College of Pharmacy & Allied Health Professions, South Dakota State University, Brookings, SD 57007, USA

**Keywords:** pharmacogenomics, *CYP2D6*, beta blocker, phenoconversion

## Abstract

Metoprolol is a medication commonly utilized in select patients to achieve a reduction in heart rate, systolic blood pressure, or other indications. A majority of metoprolol metabolism occurs via *CYP2D6*. Decreased expression of the CYP2D6 enzyme increases the concentration of metoprolol. Current pharmacogenomics guidelines by the Dutch Pharmacogenomics Working Group recommend slower titrations and dose decreases to minimize adverse effects from poor metabolizers or normal metabolizers taking concomitant medications that are strong inhibitors of CYP2D6 (phenoconverters). This study aimed to evaluate adverse effects such as bradycardia, hypotension, and syncope in patients who are expected to have absent CYP2D6 enzyme activity due to drug–drug or drug–gene interactions. The secondary aims of this study were to evaluate heart rate measurements for the included participants. Retrospective data were collected for individuals with *CYP2D6* genotyping results obtained for clinical purposes. Three categories (CYP2D6 normal metabolizers, poor metabolizers, and phenoconverters) were assigned. A total of 325 participants were included. There was no statistically significant difference found in the primary composite outcome between the three metabolizer groups (*p* = 0.054). However, a statistically significant difference was identified in the incidences of bradycardia between the poor metabolizers and the normal metabolizers or phenoconverters (*p* < 0.0001). The average heart rates were 2.8 beats per minute (bpm) and 2.6 bpm lower for the poor metabolizer and phenoconverter groups, respectively, compared to the normal metabolizers (*p* < 0.0001 for both comparisons). This study further supports the role of genetic testing in precision medicine to help individualize patient care as CYP2D6 poor metabolizers taking metoprolol were found to have an increase in bradycardia. Additional research is needed to clarify the dose relationship in this drug–gene interaction.

## 1. Introduction

Metoprolol is a commonly utilized medication for conditions such as heart failure, atrial fibrillation, and hypertension. The mechanism of metoprolol is a cardioselective, beta-1 adrenergic receptor blocker which has the desired effects of reducing heart rate and blood pressure and increasing cardiac output. The metabolism of metoprolol occurs primarily via the CYP2D6 enzyme, with reported estimates of approximately 70–80% [1]. It has been postulated that the reduction in metoprolol metabolism leads to increased concentrations and subsequently increased adverse effects associated with the medication [2,3]. A standardization of the *CYP2D6* genotype–phenotype association established four phenotype groups: ultra-rapid metabolizers (UM), normal metabolizers (NM), intermediate metabolizers (IM), and poor metabolizers (PM) (Table 1). As CYP2D6 PMs have no enzyme activity, individuals in this phenotype group are expected to exhibit a several-fold increase in metoprolol concentrations compared to NMs [3,4]. It is estimated that 8% of Caucasians are PMs, while the incidence in other populations is approximately 2% [5].

Phenoconversion is a discrepancy in the patient’s genotypic prediction of drug metabolism and the true phenotypic capacity the individual demonstrates on the metabolism of a drug; in other words, it is a drug–drug interaction due to inhibition of the CYP2D6 enzyme [7,8]. Strong inhibitors of CYP2D6 decrease the enzyme activity to near zero, causing individuals who display normal or intermediate genotypes to mimic a poor metabolizer [9]. Commonly prescribed medications that are known strong inhibitors of the CYP2D6 enzyme include bupropion, fluoxetine, paroxetine, quinidine, and terbinafine [10]. 

There is mounting evidence implying an increased risk of adverse hemodynamic outcomes due to drug–drug interactions between CYP2D6 inhibitors and beta blockers [11,12,13,14]. The corollary drug–gene interaction (e.g., using metoprolol in CYP2D6 poor metabolizers) has gained interest in the past decade, and an association between the CYP2D6 metabolizer phenotype and metoprolol serum concentrations seems likely [4,15,16,17,18,19]. Several studies have also noted a relationship between CYP2D6 phenotype and clinical outcomes and, in general, a decreasing function of CYP2D6 is associated with lower blood pressure (BP) [20,21] and a decreased heart rate (HR) [20,21,22]. Despite sound evidence on pharmacokinetic and surrogate outcomes in the literature, an association between CYP2D6 phenotype and patient-reported clinical symptoms in metoprolol users remains uncertain as studies have found both positive [23,24,25,26] and negative [27,28,29,30] associations. The Dutch Pharmacogenomics Working Group (DPWG) has published guidelines with clearly delineated recommendations for *CYP2D6* [2]. For PMs, DPWG guidelines recommend increasing the metoprolol dose in smaller steps and/or prescribing no more than 25% of the standard dose (i.e., a 75% dose reduction). The Clinical Pharmacogenetics Implementation Consortium (CPIC) has not published a pharmacogenomics (PGx) guideline for beta blockers, though one is reported to be in progress.

The PGx service at Sanford Health (Sanford Imagenetics) was created in 2014 with the goal of merging PGx into primary care. Our implementation efforts are described in depth elsewhere [31]. In brief, our approach includes both pre-emptive population screening and reactive testing capabilities and, to date, we have completed preemptive PGx panels on more than 19,000 patients. The current iteration of the PGx panel consists of 11 genes: *CYP2C9*, *CYP2C19*, *CYP2D6*, *CYP2C* cluster, *CYP3A5*, *CYP4F2*, *DPYD*, *IFNL3*, *SLCO1B1*, *TPMT*, and *VKORC1*. The metoprolol-*CYP2D6* interaction is commonly encountered in our practice; we have encountered over 500 patients with a relevant drug-gene (*CYP2D6*) or drug–drug (phenoconversion) interaction. Anecdotally, we have noticed bradycardia seems more common in our patients that are CYP2D6 PMs or those taking phenoconverting medications. PGx population studies tend to be statistically limited by the relative infrequency of the minor allele frequency. Thus, we sought to utilize the size of our population to assess the clinical outcomes. Therefore, the aim of this study was to systematically evaluate the effect of the CYP2D6 phenotype on the side effects of metoprolol in a cohort of our patients.

## 2. Materials and Methods

This was a retrospective, single-health-system, cohort study. Participants were included in the trial if they were older or equal to 18 years of age, had documented results of the *CYP2D6* genotype and phenotype within their electronic medical record (EMR), and had a history of at least one outpatient metoprolol prescription. Exclusion criteria included a documented pacemaker, primary bradycardia, which was defined as a heart rate (HR) less than 55 beats per minute (bpm), or concomitant use of diltiazem, verapamil, or digoxin while on metoprolol (ivabradine was not formally excluded, but no study participants were found to be taking this medication). Participants were chosen initially from the PM population and were matched 1:1 to NMs based on age, sex, and race. The phenoconverter cohort was subsequently matched to the NMs and PMs in the same way. Matching was performed using simple random selection. The primary outcome was a composite incidence of symptomatic bradycardia (i.e., HR < 60 bpm with documentation of adverse reaction), hypotension (i.e., systolic blood pressure (SBP) < 90 or diastolic blood pressure (DBP) < 60), and syncope (i.e., a documented loss of consciousness associated with low blood pressure). Secondary outcomes included the individual components of the primary outcome, incidence of patient-self-reported lightheaded and dizziness, and heart rates for the included patients. 

The outcome variables were collected via manual chart review of progress notes searching for the key terms (bradycardia, hypotension, syncope, dizziness, or lightheadedness) and if there was a concurrent active metoprolol prescription. If the variable was documented in the patient’s chart, it was confirmed against the medication list to ensure overlap with an active metoprolol prescription or documentation within the provider’s notes that the patient was actively taking metoprolol. The overlap of the metoprolol and phenoconverting medications was also verified based on the medication list in the patients’ charts. Patients were classified as phenoconverters based on a documented normal CYP2D6 phenotype with a documented overlap of metoprolol and one of the three following medications: bupropion, fluoxetine, or paroxetine (no individuals in the study population were taking quinidine or terbinafine). Outcome variables were only collected during the time of phenoconversion overlap. Notably, we limited the phenoconverter group to strong inhibitors of CYP2D6 only and identified phenoconverters only from the NM group. This was done with the intention of creating three groups that avoided confounding drug–drug and drug–gene interactions. 

Hemodynamic data—HR, SBP, and DBP—were collected from the EMR using an automated data query. The analysis focused on HR due to the significant secondary outcome in bradycardia. We collected HR measurements for all ambulatory patient visits taken throughout the study observation period. The average heart rate and minimum heart rate (i.e., an average of the lowest recorded heart rate) were analyzed using the Kruskal –Wallis test, and Dunn’s test was used for a post-hoc pairwise comparison. 

An estimated 2500 patients fit the inclusion criteria. Previous studies found a rate of 9% of symptomatic bradycardia in PMs and 1.6% in NMs, suggesting an effect size of 0.122 using Cramers V method [30]. Assuming two metabolizer groups and a phenoconverter group, a sample size of 145 participants in each group was needed to produce an 80% power to detect a significant outcome. Chi square testing was utilized for a test of difference between the three metabolizer groups. Multivariate logistic regression was used to identify predictors of the outcomes. The model included covariates of age, sex, and race, using a 95% Wald confidence interval for significance. The Shapiro–Wilk test defined the heart rate data as nonparametric, allowing for test of difference to be performed using the Kruskal–Wallis test. 

*CYP2D6* genotyping was performed at the Sanford Medical Genetics Laboratory (SMGL), which utilizes multiple platforms to ascertain *CYP2D6* genotype and phenotype. The following alleles were tested for *CYP2D6*: *2, *3, *4, *4M, *5 (deletion), *6, *9, *10, *17, *29, and *41. Fluidigm SNP Dynamic Array [Standard BioTools; South San Francisco, CA] is utilized for genotyping; copy number assessment is conducted via droplet digital PCR [Bio-Rad; Hercules, CA, USA]. 

## 3. Results

There were 189 PMs who met the inclusion criteria for this study; however, after accounting for exclusion criteria, 114 PMs were included in the final analysis. These were paired 1:1 with an age-, sex-, and race-matched individual who was selected at random from the NM group, accounting for the 114 individuals in the NM group. The phenoconverter group was likewise matched to the PM group, though only 97 individuals were found to have a qualifying drug–drug interaction (Figure 1).

Baseline characteristics for the study population, which was predominantly Caucasian and had a nearly equal split between male (48.6%) and female (51.4%) participants overall, are provided in Table 2. The average age was 66.8, 66.9, and 65.2 years old in the NM, PM, and phenoconverter groups, respectively. The study cohort demographics largely matched the demographics in the region we serve. Given the retrospective nature of this study, there was a small numeric variation in the baseline characteristics, though there were no statistically significant differences between phenotype groups. A total of 325 patients were included in the analysis: NM = 114, PM = 114, and phenoconverters = 97. 

Figure 2 describes the composite incidence of bradycardia, hypotension, and syncope in the three study groups; individual components of the primary are also depicted. There was a numeric increase in events in the PM group (50%) vs. the NM group (34%) but less so in the phenoconverter group (39.5%). Statistical significance was not achieved for the primary outcome (*p* = 0.054). Among secondary outcomes, a statistically significant difference was noted in the bradycardia outcome between the three groups (*p* < 0.0001). The incidence of bradycardia was 25.4%, 41.2%, and 14.4% in the NMs, PMs, and phenoconverter groups, respectively. There were no other differences found amongst the secondary outcomes. A regression analysis (Table 3) was completed to account for predefined variables of age, sex, and race. The regression analysis showed an association between bradycardia and PM phenotype. The odds ratio for bradycardia between the NM and PM groups was 1.62 (95% CI 1.11 to 2.39; *p* < 0.0001). Age was a notable driver for this association (*p* = 0.0002). Figure 3 highlights a descriptive visual for the incidence of bradycardia stratified by age. While the overall trend shows the incidence of bradycardia increases with age, PMs are numerically the most affected, especially in age groups greater than 65 years old. 

HR was a predefined secondary outcome of this study due to the drug’s mechanism of action. Figure 4 shows the average and minimum heart rates recorded for the study participants. The average heart rates (Figure 4a) were 75.9 ± 16.8 bpm, 73.1 ± 15.5 bpm, and 73.3 ± 13.8 bpm in the NMs, PMs, and phenoconverters, respectively (*p* < 0.0001). Pairwise comparisons between NMs vs. PMs (*p* < 0.0001) and NMs vs. phenoconverters (*p* < 0.0001) were statistically significant but were non-significant between PMs and phenoconverters. The minimum heart rate (Figure 4b) was found to be statistically different between the three metabolizer groups (*p* = 0.0052). The mean minimum HRs were 62.5 ± 10.8 bpm, 58.0 ± 10.3 bpm, and 61.8 ± 9.5 bpm for NMs, PMs, and phenoconverters, respectively. While the study groups were relatively equal in size, the PM group had more BP assessments documented (4053 BP assessments) compared to phenoconverters (2670 BP assessments) and NMs (2422 BP assessments). This finding was not included in the overall purpose of study, but is an interesting finding noted through data analysis.

## 4. Discussion

The results of our study indicate that CYP2D6 PMs have an increased risk of bradycardia and a lower minimum HR compared to NMs and phenoconverters. The incidence of bradycardia was 62% higher in the PM group vs. the NM group, and the minimum HR was four bpm lower in the PM group compared to the NM and phenoconverter groups. The primary composite outcome—incidence of bradycardia, hypotension, and syncope—was not statistically different between groups, though the study was initially powered to find a difference in the bradycardia outcome alone. The phenoconverter group demonstrated no meaningful differences in clinical outcomes when compared to the NM group. This was unexpected but is possibly related to the small cohort size and the non-standardized time of phenoconversion overlap.

This study contributes to the growing body of evidence for the relationship between *CYP2D6* and metoprolol. There exists substantial literature related to metoprolol PGx, though the evidence is more compelling for surrogate outcomes (e.g., HR and BP) than clinical outcomes. A genetic substudy of the MERIT-HF trial found a greater decrease in HR and DBP outcomes for CYP2D6 IM and PM groups compared to the NM group [20]. This effect was most pronounced during early titration and when reaching target dose (i.e., 200 mg of metoprolol succinate daily), thus suggesting a dose-response effect. A prospective study by Thomas and colleagues found that a lower CYP2D6 activity score was significantly associated with a greater reduction in HR (*p* < 0.001) [22]. This study, which was performed in the USA, provided a demographically diverse (>30% African American, ~50% female) population, and the HR reduction in the PM group was nearly twice that of the NM group (−13.7 bpm vs. −7.1 bpm, respectively), though the association was non-significant. An observational study in Germany by Wuttke and colleagues found a fivefold higher risk for metoprolol-related adverse effects in CYP2D6 PMs vs. non-PMs in an at-large population [23]. A Dutch study by Bijl and colleagues found an 8.5 bpm decrease in HR, leading to increased risk for bradycardia, in CYP2D6 PMs taking metoprolol compared to CYP2D6 NMs (OR = 3.86, *p* = 0.0014) [25]. Recently, Chen and colleagues performed a prospective observational trial of metoprolol tolerance in a Han Chinese population, stratifying adverse events by CYP2D6 phenotype [26]. While few PMs were enrolled in the study, the study did find a statistically higher incidence of postural hypotension, bradycardia, asystole, and syncope in the IM group compared to the NM group. The overall incidence of cardiovascular adverse events for IMs was 60% higher than the NM group in Chen et al. (OR = 1.06; *p* < 0.001). Comparatively, while the composite primary outcome in the present study was non-significant, bradycardia was found to be significantly more common in the PM group (OR = 1.62; *p* < 0.0001). Taken in the context of previous studies, our results are encouraging that a clinical outcome may, in fact, exist. A prospective randomized clinical trial is likely needed to find a definitive association between the *CYP2D6* genotype and metoprolol adverse effects.

The strengths of this study include its cohort matching, broad population sampling secondary to limited exclusion criteria, and the large PGx population that was able to be included for this study. A manual chart review of adverse effects likely contributed to the high yield of incidence of outcomes instead of relying on documented ICD-9/-10 codes in the EMR. Moreover, the large sample size of the PM group (often a limiting factor in PGx studies) and the relatively balanced number of individuals in each group led to a more robust statistical analysis. Correlations among clinical and surrogate outcomes are also compelling, though the clinical significance of the observed heart rate differences is debatable.

There were several key limitations to this study. First, the study was underpowered to detect a difference in the primary outcome. Original inclusion estimates placed the number of individuals in each group around 145; however, after accounting for exclusion criteria (in the PM group) and verifying the drug–drug interactions (in the phenoconverter group), the number of individuals included in each group was significantly limited. As a result, the desired sample size was not obtained. This is particularly noteworthy as the primary outcome bordered on statistical significance. However, the effect seen in the bradycardia secondary outcome is concordant with those seen in previous studies [23,25], and we believe that the bradycardia outcome is worth reporting as it is a clear driver of the primary outcome and is clinically relevant.

Second, due to the retrospective nature of this study, we could not directly assess medication compliance related to both phenoconversion medications and overall metoprolol use. Without the ability to assess compliance, participants in the phenoconverter group may not actually have been on the inhibitor at the time of metoprolol use, which is potentially why the results were not as expected for this group. Furthermore, phenoconversion was addressed strictly as an all-or-none phenomenon in that only NMs taking strong CYP2D6 inhibitors were analyzed in this cohort. Future studies should seek to explore the effect of phenoconversion within other metabolizer groups and with moderate CYP2D6 inhibitors.

Third, our study did not specifically address metoprolol dose or formulation, only whether patients were taking metoprolol or not. Further analysis of the dose effect on the outcomes would be beneficial to help correlate a potential relationship between genotype and maintenance dose. The analysis also did not distinguish between metoprolol formulation (tartrate vs. succinate). While the clinical implications of metoprolol formulation within the context of drug-gene and drug-drug interactions are not fully understood, there are important pharmacokinetic differences that may impact drug interactions such as phenoconversion [22].

Finally, the characteristics of the study population need to be considered when applying and interpreting this study. The participants in this study were generally reflective of our rural Midwestern patient population, which tends to be older and self-identifies as being of northern European ancestry. The frequency of PMs in individuals with European ancestry is approximately 6–8% vs. 1–3% for other ethnic groups [5]. The outcomes studied in this trial may be less pronounced in non-white populations and may limit the application of these results to areas that serve more diverse populations. Additionally, the average age of the study population was 66.4 years old, and the multivariate analysis found a positive correlation between age and bradycardia. While this result is not surprising, applying these results to younger patients may overestimate the risk. Notably, the multivariate analysis identified age as an independent variable but did not model this as a confounder. Thus, the impact of age vs. genotype was not fully elucidated. Investigators may wish to consider age as an a priori confounder in future studies.

## 5. Conclusions

This study found that CYP2D6 PMs taking metoprolol had a significant increase in the incidence of symptomatic bradycardia compared to genotypic NMs. The study did not reach statistical significance surrounding the primary composite outcome, however. Nonetheless, this study supports previous findings that metoprolol may be clinically impacted by CYP2D6 phenotype, though future research is warranted to investigate the impact of a wider range of CYP2D6 phenotypes. While a statically significant difference in bradycardia between CYP2D6 NMs and PMs was found, CYP2D6 major inhibitors did not produce the same effect in this study. Additional research into the impact of CYP2D6 phenoconversion and the role drug–drug interactions play on metoprolol is also needed.

## Figures and Tables

**Figure 1 jpm-13-00416-f001:**
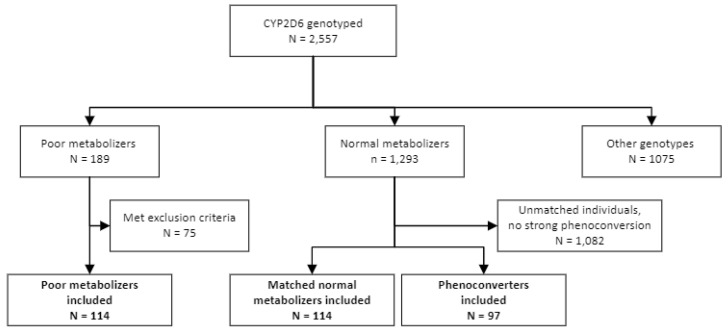
Flowchart of study participants.

**Figure 2 jpm-13-00416-f002:**
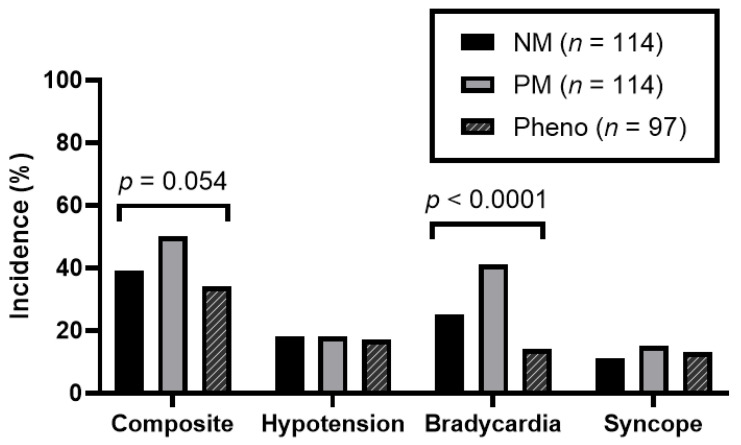
Incidence of primary composite outcome and individual components of the primary.

**Figure 3 jpm-13-00416-f003:**
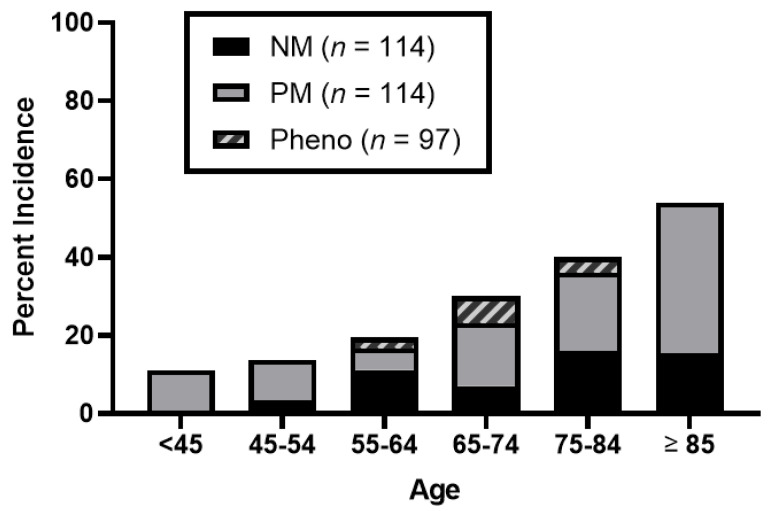
Incidence of bradycardia among CYP2D6 phenotypes stratified by age.

**Figure 4 jpm-13-00416-f004:**
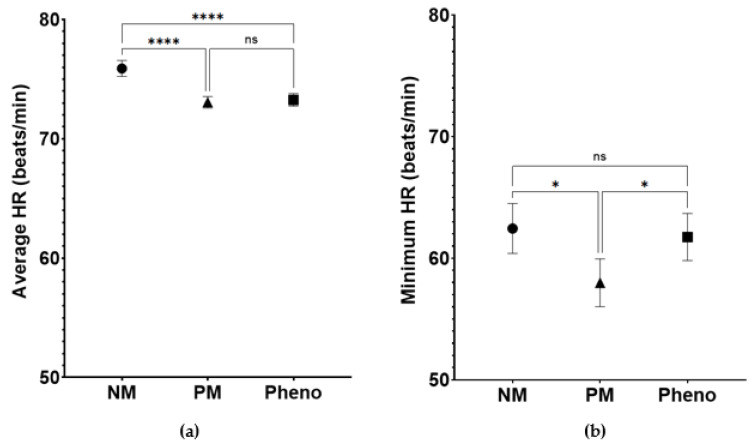
Ambulatory measurements of heart rate for all patients taken throughout the study observation period: (**a**) average of all heart rate measurements (mean +/− 95% CI) stratified by CYP2D6 phenotype group and (**b**) average minimum heart rate. NM:normal metabolizers; PM: poor metabolizers; Pheno: phenoconverter group. * *p* < 0.5; **** *p* < 0.001; NS: non-significant.

**Table 1 jpm-13-00416-t001:** Predicted *CYP2D6* phenotypes based on diplotypes.

Phenotype	Activity Score Range	Activity Score/Genotypes	Examples of *CYP2D6* Diplotypes
Ultra-rapid metabolizer	>2.25	>2.25	*1/*1×N, *1/*2×N, *2/*2×N
Normal metabolizer	1.25 < x < 2.25	1.25 1.5 1.75 2.0 2.25	*1/*10 *1/*41, *1/*9 *10/*41×3 *1/*1, *1/*2 *2×2/*10
Intermediate metabolizer	0 < x < 1.25	0.25 0.5 0.75 1	*4/*10 *4/*41, *10/*10 *10/*41 *41/*41, *1/*5
Poor metabolizer	0	0	*3/*4, *4/*4, *5/*5, *5/*6

Table adapted from Caudle et al. (2020) [6].

**Table 2 jpm-13-00416-t002:** Baseline characteristics of the study population.

	Normal Metabolizer	Poor Metabolizer	Phenoconverter
Demographic	*n* = 114	*n* = 114	*n* = 97
**Age (yrs.)**			
<55	16 (14%)	16 (14%)	15 (15%)
55–64	24 (21%)	24 (21%)	24 (25%)
65–74	50 (44%)	51 (45%)	42 (43%)
75+	24 (21%)	23 (20%)	16 (16%)
**Sex**			
Male	58 (51%)	58 (51%)	42 (43%)
Female	56 (49%)	56 (49%)	55 (57%)
**Race**			
Caucasian/white	114 (100%)	113 (99%)	96 (99%)
American Indian/Alaskan Native	0	1 (1%)	0
Asian	0	0	1 (1%)
**Concomitant Medications**			
Bupropion	0	n/a	59 (61%) ^
Fluoxetine	0	n/a	40 (41%) ^
Paroxetine	0	n/a	16 (16%) ^

n/a = not assessed; ^ Concomitant Medications percentages equal greater than 100% due to overlap of multiple medications.

**Table 3 jpm-13-00416-t003:** Incidence of primary and secondary outcomes stratified by CYP2D6 phenotype.

	Event Incidence (%)			Odds Ratio (95% CI)	
Outcome	NM (*n* = 114)	PM (*n* = 114)	Pheno (*n* = 97)	NM vs. PM	NM vs. Pheno
Primary Composite	45 (39.5)	57 (50)	33 (34)	1.27 (0.95 to 1.70)	0.862 (0.599 to 1.23)
Individual Components of Primary					
Hypotension	21 (18.4)	21 (18.4)	17 (17.5)	1.00 (0.58 to 1.72)	0.95 (0.54 to 1.68)
Bradycardia	29 (25.4)	47 (41.2)	14 (14.4)	1.62 (1.11 to 2.39) *	0.57 (0.32 to 1.00)
Syncope	13 (11.4)	18 (15.8)	13 (13.4)	1.39 (0.72 to 2.67)	1.18 (0.58 to 2.38)
Additional Secondary Outcomes					
Lightheadedness	39 (34.2)	44 (38.6)	33 (34)	1.13 (0.80 to 1.59)	0.99 (0.68 to 1.4)
Dizziness	59 (51.8)	62 (54.4)	46 (47.4)	1.05 (0.82 to 1.35)	0.92 (0.69 to 1.20)

* *p* < 0.0001; NM: normal metabolizer; PM: poor metabolizer; Pheno: phenoconverter.

## Data Availability

Not applicable.

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
