# Peer review of "Metoprolol and CYP2D6: A Retrospective Cohort Study Evaluating Genotype-Based Outcomes"

_jpm, 2023, doi:10.3390/jpm13030416_

Round 1
Reviewer 1 Report
The study is interesting. It shows a statistically significant difference in the incidence of bradycardia and heart rate between the poor metabolizers and the normal metabolizers or phenoconverter groups.
Abstract: Please include the heart rate results where the difference was statistically significant between the normal and poor or phenoconverter groups.
Line 166: Incidences of bradycardia were 41.2%, 14.4%, and 25.4% in the PM, NM, and phenoconverter groups, respectively. Based on Figure 2, I believe there is a mistake, the incidence of bradycardia in the NM is 25.4% and 14.4% in the phenoconverter group.
Figure 3: X axis, please fix the symbol: > 85
Please restructure the discussion, highlight first the positive results and their importance, then discuss the limitations of the study and future studies. For example, the last paragraph may be the second paragraph.
Discussion: Please include more citations and compare the results to the ones obtained in other countries for all the studied components.
Thank you
Author Response
Response to Reviewer 1 Comments
Thank you for the constructive and collegial review of our manuscript. The manuscript document has been revised as requested, making an attempt to address each of your comments as thoroughly as possible. Please find a point-by-point response (in red) to your comments listed below:
Comment 1 - Abstract: Please include the heart rate results where the difference was statistically significant between the normal and poor or phenoconverter groups.
Blood pressure results have been added to the abstract as requested. For clarity, we chose to describe the average heart rate data only and to focus on the relative decrease within this metric.
Comment 2 - Line 166: Incidences of bradycardia were 41.2%, 14.4%, and 25.4% in the PM, NM, and phenoconverter groups, respectively. Based on Figure 2, I believe there is a mistake, the incidence of bradycardia in the NM is 25.4% and 14.4% in the phenoconverter group.
Thank you for correcting this error. The text in this sentence has been revised to correct the error and to match the order of the data in Table 3. The sentence will now read: “Incidence of bradycardia were 25.4%, 41.2%, and 14.4% in the NM, PM, and phenoconverter groups, respectively.”
Comment 3 - Figure 3: X axis, please fix the symbol: > 85
Figure 3 has been updated to correctly display the “greater than or equal” symbol in the X-axis.
Comment 4 - Please restructure the discussion, highlight first the positive results and their importance, then discuss the limitations of the study and future studies. For example, the last paragraph may be the second paragraph.
Comment 5 - Discussion: Please include more citations and compare the results to the ones obtained in other countries for all the studied components.
Addressing the concerns in Comments 4 and 5, we have revised and restructured the discussion section as recommended. The last paragraph was moved up to the second paragraph, and we added discussion about previous studies that relate the various outcomes addressed in our study (HR, bradycardia). We also added discussion of a recent study from a Han Chinese population (Front Pharmacol. 2022 Apr 6;13:876392) that further supports the association between genotype and clinical outcomes. We also made an effort to identify the countries where different studies were conducted to allude to diversity of results.
Reviewer 2 Report
This article studied the adverse effect of metoprolol when it was administered to patients with inactive/ low activity of CYP2D6. Since a significant number of patients use this beta-blocker, this study's aim is attractive in Pharmacogenomics.
It could be better if the co-administration of CYP2D6 inhibitors (drugs) were studied. Also, the different race has different phenotype distributions, and there is some room for study in the future, as the authors mentioned in the article.
This manuscript is a revision, and I can see a lot of improvement from the original. Yet, there are some limitations, as the other reviewers mentioned. The results are still valuable since the information in this manuscript is beneficial for further considerations in clinical study and the evaluations of different drugs in pharmacogenomics.
I recommend publishing this manuscript after minor revisions in the Journal of Personal Medicine. I hope they continue to do further study.
In Figure 2 and 3, legend: PM (n = 114 ïƒ PM (n = 114 ) Need closing parenthesis.
Author Response
Response to Reviewer 2 Comments
We appreciate the constructive and collegial review of our manuscript. The comments from Reviewer 2 generally highlight limitations in the study design, which we have addressed in previous revisions. The present revision does not make further changes to design.
Please find a point-by-point response below to the reviewer's specific comments:
Comment 1 - In Figure 2 and 3, legend: PM (n = 114 --> PM (n = 114) Need closing parenthesis.
The legends in Figure 2 and 3 have been updated as requested. Thank you for catching this error.